# Impact of a Prenatal Vitamin D Supplementation Program on Vitamin D Deficiency, Rickets and Early Childhood Caries in an Alaska Native Population

**DOI:** 10.3390/nu14193935

**Published:** 2022-09-22

**Authors:** Rosalyn J. Singleton, Gretchen M. Day, Timothy K. Thomas, Joseph A. Klejka, Christine A. Desnoyers, Melanie N. P. McIntyre, David M. Compton, Kenneth E. Thummel, Robert J. Schroth, Leanne M. Ward, Dane C. Lenaker, Rachel K. Lescher, Joseph B. McLaughlin

**Affiliations:** 1Department of Research Services, Alaska Native Tribal Health Consortium, Anchorage, AK 99508, USA; 2Department of Quality, Yukon Kuskokwim Health Corporation, Bethel, AK 99559, USA; 3Laboratory, Yukon Kuskokwim Health Corporation, Bethel, AK 99559, USA; 4Department of Obstetrics and Gynecology, Yukon Kuskokwim Health Corporation, Bethel, AK 99559, USA; 5School of Pharmacy, University of Washington, Seattle, WA 99559, USA; 6Department of Preventive Dental Science, University of Manitoba, Winnipeg, MB R3E 3P4, Canada; 7Department of Pediatrics and Child Health, University of Manitoba, Winnipeg, MB R3E 3P4, Canada; 8Department of Pediatrics, University of Ottawa, Ottawa, ON K1 H8 L1, Canada; 9Department of Dentistry, Southeast Alaska Regional Health Consortium, Juneau, AK 99801, USA; 10Department of Pediatrics, Alaska Native Tribal Health Consortium, Anchorage, AK 99508, USA; 11Division of Public Health, State of Alaska, Anchorage, AK 99508, USA

**Keywords:** Vitamin D, rickets, pregnancy, caries, early childhood caries, vitamin D supplementation, vitamin D deficiency

## Abstract

Background: Early childhood rickets increased in Alaska Native children after decreases in vitamin D-rich subsistence diet in childbearing-aged women. We evaluated the impact of routine prenatal vitamin D supplementation initiated in Alaska’s Yukon Kuskokwim Delta in Fall 2016. Methods: We queried electronic health records of prenatal women with 25(OH) vitamin D testing during the period 2015–2019. We evaluated 25(OH)D concentrations, vitamin D_3_ supplement refills, and decayed, missing, and filled teeth (dmft) scores and rickets in offspring. Results: Mean 25(OH)D concentrations increased 36.5% from pre- to post-supplementation; the percentage with deficient 25(OH)D decreased by 66.4%. Women with ≥ 60 vitamin D_3_ refill days had higher late pregnancy 25(OH)D concentrations than those with no refill days (*p* < 0.0001). Women with late pregnancy insufficient 25(OH)D concentrations had offspring with higher dmft scores than those with sufficient 25(OH)D (RR 1.3, *p* < 0.0001). Three children were diagnosed with nutritional rickets during the period 2001–2021, and none after 2017. Conclusions: These findings suggest that prenatal vitamin D supplementation can improve childhood outcomes in high-risk populations with high rates of rickets.

## 1. Introduction

People living in northern latitudes are at increased risk of vitamin D deficiency due to limited ultraviolet B exposure and diminishing dietary intake of vitamin D-rich traditional foods. As a result, the incidence of rickets has increased in some northern latitude populations [1,2,3]. During the period 2001–2010, Alaska Native children aged <10 years experienced nearly twice the rate of rickets-associated hospitalization compared to other U.S. children aged <10 years (2.23 vs. 1.23 cases per 100,000). [3]. The incidence of rickets was 4.2/100,000 [3], and 63% were aged <1 year. Alaska Native children with rickets/vitamin D deficiency had similar breast-feeding prevalence but were less likely to have received vitamin D supplementation. Alaska Native children continued to be diagnosed with rickets in Alaska during the period 2011–2016 despite education on infant vitamin D supplementation, and case studies from Canada demonstrated that rickets could occur in supplemented infants born to mothers with severe vitamin D deficiency [4].

Historical dietary evaluations of Alaska Native people have shown higher vitamin D intake in those eating a traditional subsistence diet [5,6]; however, traditional food consumption has decreased among Alaska Native people from the 1980s to present [5,7,8,9,10]. The National Academy of Medicine considers a 25-hydroxyvitamin D (25(OH)D) concentration ≥20 ng/mL (≥50 nmol/L) sufficient for bone and overall health, while <12 ng/mL (<30 nmol/L) is considered deficient and is associated with increased risk of developing rickets [11,12,13,14]. The historical role of diet on vitamin D levels was evaluated in childbearing women from Alaska’s Yukon Kuskokwim (YK) Delta. Intake of a traditional marine diet, as measured by serum Nitrogen Isotope Ratio values, decreased linearly by decade during the period 1960–1999 [15]. Serum 25(OH)D concentrations were significantly related to Nitrogen Isotope Ratio values, and the proportion of women with sufficient 25(OH)D levels decreased from 100% in the 1960s-1970s to 53% in the 2010s (*p* < 0.0001). 

Exposure to sugars, improper infant feeding, inadequate dental hygiene and lack of access to care are known risk factors for early dental decay. Vitamin D insufficiency/deficiency can adversely affect tooth mineralization of primary and permanent dentition [16,17,18]. A cohort study in a Canadian urban prenatal population demonstrated that lower prenatal 25(OH)D levels were significantly associated with higher numbers of decayed primary teeth in their offspring [19]. Alaska Native children experience one of the highest reported rates of early childhood caries (ECC), resulting in costly restorations and pain [20]. In Alaska, we examined 25(OH)D in cord blood and ECC for YK Delta mother/infant pairs participating in the Maternal Organic Monitoring study [21]. Children aged 12–35 months with deficient cord blood 25(OH)D levels had a mean decayed missing and filled teeth (dmft) score twice as high as children who were nondeficient [22]. Among study participants, 28% of prenatal blood samples and 91% of cord blood samples had insufficient 25(OH)D concentrations [23]. 

Prompted by the high prevalence of prenatal vitamin D insufficiency in the Maternal Organic Monitoring study, the YK Health Corporation (YKHC) tribal health organization evaluated 25(OH)D in women at delivery during summer 2016 and determined that 60% (24/40) had 25(OH)D insufficient concentrations. Since prenatal vitamin D deficiency can contribute to infant rickets, in September 2016, the YKHC medical director signed guidelines and standing orders implementing routine prenatal vitamin D_3_ supplementation with 25 µg (1000 International Units [IU]) of daily vitamin D_3_ in addition to the vitamin D in routine prenatal vitamins and calcium supplementation (600 IU/day). YKHC instituted a period of routine prenatal 25(OH)D screening at the first prenatal visit and late pregnancy to evaluate the impact of this new program. Here, we present data on the acceptability and impact of the YKHC prenatal vitamin D supplementation program on maternal 25(OH)D concentrations, ECC, and rickets risk in their offspring.

In 2017, the Alaska Division of Public Health created a Vitamin D Workgroup to review the scientific evidence of the effects of vitamin D on human health and evaluate the need for Alaska-specific vitamin D recommendations for diet and supplementation. In 2018, the workgroup published Alaska-specific vitamin D supplementation recommendations for prenatal women and infants [23,24]. These statewide recommendations for prenatal women aligned with the 2016 YK Delta recommendations: supplement with 1000 IU/day in addition to daily prenatal vitamins. In addition, 400 IU/day was recommended for all Alaskan infants irrespective of feeding mode, with an additional 400 IU/day advised for those who were partially or completely breastfed. 

## 2. Materials and Methods

### 2.1. Setting

This study focused on prenatal Alaska Native women and their offspring in the YK Delta region, which encompasses 75,000 square miles of southwestern Alaska. The region’s population of approximately 27,000 is comprised primarily of Yup’ik Alaska Native people who live in 52 small village communities and the regional hub town of Bethel. Pre-paid healthcare implemented under the federal Indian Health Service is provided to all Alaska Native people through YKHC at the YK Delta Regional Hospital and at primary care clinics in YK Delta village communities. Patients requiring tertiary care are transported to Alaska Native Medical Center (ANMC) in Anchorage. The YK Delta communities are connected by air, water, and snowmobile, with no road access to the remainder of Alaska. One-third of the communities do not have piped water. Among those communities with piped water, only Bethel was fluoridating water at the time of this report.

This study was approved by the Alaska Area Institutional Review Board, and the board of directors for the tribal health organizations, YKHC and Alaska Native Tribal Health Consortium, with a waiver of informed consent.

### 2.2. Study Design 

In this cohort study, we queried the YKHC and ANMC electronic health records of all YK Delta prenatal women with 25(OH)D testing during the period, June 2015–December 2019 (175 pre-supplementation (2015–2016) and 1347 post-supplementation (2017–2019)). Abstracted data included prenatal visits and birth records, plasma 25(OH)D concentrations, maternal age (<33 years vs. ≥33 years), subregion (coastal vs. river community), parity, maternal tobacco use, and season (Nov–Apr vs. May–Oct) at the time of 25(OH)D test. Covariates were chosen based on their published or potential association with prenatal 25(OH)D levels [5,15]. We defined vitamin D sufficiency as ≥20 ng/mL (50 nmol/L), insufficiency as ≥12 ng/mL (30 nmol/L) and <20 ng/mL (50 nmol/L), and deficiency as <12 ng/mL (30 nmol/L) [25]. YKHC pharmacy electronic records were queried for vitamin D_3_ supplement refills after initial prescription for a subset (deliveries in 2018) of prenatal women.

We queried the electronic health and electronic dental records of infants born to the prenatal women in our study during the first 35 months of life from January 2015 through December 2020. The first 35 months of life were chosen because of prior data showing an association between cord blood 25(OH)D and dmft for this age range [22]. We generated dmft scores (cumulative number of primary teeth decayed, missing due to caries, or filled) from the YKHC electronic dental records by age in days at exam. Covariates with potential relationship to dmft were assessed, including gestational age at delivery, birth weight, presence of dental provider in the community, and maternal smoking. 

To evaluate the trend in confirmed nutritional rickets during the period 2001–2021, the pediatric endocrinologist evaluated the clinical and laboratory findings for potential cases of rickets (ICD9 268.0, 268.1 or ICD10 E55.0) in Alaska Native children aged 0–9 years identified in the electronic medical record, as previously published [3]. Children were included as having nutritional rickets if they had clinical/radiologic evidence of rickets, not associated with underlying malabsorption or hepatic disease. 

### 2.3. Data Analysis

We compared mean 25(OH)D concentrations in late pregnancy (20+ weeks gestation) of YK Delta prenatal women post-supplementation (2017–2018) to controls pre-supplementation (2015–2016). Significant differences in means were determined using an independent *t*-test. We also compared the proportions of the pre-supplementation and post-supplementation cohorts whose 25(OH)D concentrations were sufficient, insufficient, or deficient, using a chi-square test of independence. We assessed the relationship between maternal factors and 25(OH)D concentrations at 20+ weeks gestation using an independent *t*-test. We used multiple linear regression to test which maternal factors significantly predicted 25(OH)D concentration. We examined changes in vitamin D concentration for prenatal women with a 25(OH)D value in both early (6–19 weeks) gestation and late (20+ weeks) gestation to determine whether vitamin D increases were due to supplementation or other factors. We compared changes in mean values using a paired *t*-test. We also compared early pregnancy 25(OH)D concentrations between prenatal women pre-supplementation with women post-supplementation to determine whether pre-supplementation vitamin D concentrations were representative of prenatal women from the region. Mean values were compared using independent *t*-tests. 

We determined adherence to vitamin D supplementation among a subgroup of prenatal women (n = 351) in the post-supplementation cohort. A one-way ANOVA was performed to compare the effect of the number of 30-day pharmacy refills a woman requested after her initial vitamin D prescription on maternal 25(OH)D concentrations in late pregnancy. We used Tukey’s HSD test to determine which of the multiple comparisons were significant.

We examined the effect of 25(OH)D concentrations on children’s dmft scores. We conducted a negative binomial regression using dmft scores as the dependent variable, and insufficient/deficient vs. sufficient late pregnancy vitamin D concentrations as the independent variable. Age in days was included as an offset. We restricted the analysis to children with age in days from 365 to 1095 (children aged 12–35 months). We tested gestational age, maternal age, parity, birthweight, access to dental health aide therapists/dentists, and maternal smoking as independent factors affecting dmft development in addition to 25(OH)D. We also performed a logistic regression to examine the relationship of maternal and community characteristics on the probability of ECC (any caries >12 and <36 months of age). 

We calculated incidence rates per 100,000 children aged 0–9 years for rickets (ICD9 268; ICD10 E55.0, using cases as the numerator and annual population estimates for Alaska Native children aged 0–9 years multiplied by 100,000 as the denominator) [26]. Rates for years 2001–2016 were compared to rates for 2017–2020 using a chi square test of independence. All analyses were conducted using R software (R Core Team 2019). 

## 3. Results

### 3.1. Prenatal 25(OH)D Concentrations 

A total of 1522 prenatal women with late pregnancy plasma 25(OH)D concentrations were included in the study, 175 pre-supplementation (2015–2016) and 1347 post-supplementation (2017–2019). The mean maternal age was 25.9 years (SD 5.6, range, 12–47) and the mean gestational age at the time of 25(OH)D measurement was 33.1 weeks (SD 5.9) (Table 1). 25(OH)D was higher in May–October than November–April (27.1 (SD 9.7) vs. 25.9 (SD 11.4, *p* = 0.0003) and higher in river than coastal communities (27.3 [SD 10.0] vs. 25.5 [SD 11.2], *p* = 0.0003) (Table 2) but was not associated with maternal or gestational age in late pregnancy. 

The mean late pregnancy 25(OH)D concentration increased by 36.5%, from 20.0 ng/mL pre-supplementation (2015–2016) to 27.3 ng/mL post-supplementation (2017–2019); *p* < 0.0001). During these periods, the proportion of late prenatal women with deficient (<12 ng/mL) 25(OH)D decreased by 66.4% to 5%, and the proportion with insufficient levels (12 to <20 ng/mL) decreased by 54.1% (*p* < 0.0001) (Table 3, Figure 1). There were 32 prenatal women with 25(OH)D concentrations >50 ng/mL during the post-supplementation period, and the maximum 25(OH)D concentration was 100 ng/mL (Table 3). 

During the 2017–2019 supplementation period, among the 1095 prenatal women with 25(OH)D concentrations in both early (6–19 weeks gestation) and late (20+ weeks gestation) pregnancy, there was a significant increase in mean 25(OH)D concentration from 21.1 ng/mL (SD 7.5) at 6–19 weeks gestation to 27.9 ng/mL (SD 10.3) (*p* < 0.0001) at 20+ weeks gestation. In contrast, during the pre-supplementation period (2015–2016), there was no significant change in 25(OH)D between early and late pregnancy (20.4 ng/mL vs. 20.9 ng/mL, *p* = 0.73, n = 50). There was also no significant difference between mean early pregnancy 25(OH)D concentrations for the pre-supplementation and post-supplementation groups (20.4 ng/mL [SD 7.2] vs. 21.1 ng/mL [SD 7.5], *p* =0.52).

We evaluated the relationship between vitamin D_3_ supplementation refills and 25(OH)D at ≥36 weeks gestation for a subset of prenatal women in our cohort from 2018 (n = 351). Mean plasma 25(OH)D was 35.7 ng/mL in women (n = 54) who received >120 days of vitamin D_3_ refills versus 24.3 ng/mL for women (n = 106) who received 0 days of refills (*p* < 0.0001). Tukey’s honest significant difference (HSD) test for multiple comparisons found that women who received at least two refills (60 days) had significantly higher 25(OH)D than those who had no refills (no refills vs. two refills, *p* = 0.05; no refills vs. three refills, *p* = 0.0002).

### 3.2. Birthweight and Prematurity

The mean birth weight was 3.54 kilograms (SD = 0.49 kg) among 1266 prenatal women whose infant’s birthweight was reported. There was no significant difference in the mean birth weight between women with deficient, insufficient, or sufficient 25(OH)D concentrations at ≥36 weeks gestation (*p* = 0.77). 

### 3.3. Early Childhood Caries

Prenatal women with insufficient/deficient 25(OH)D concentrations at ≥36 weeks gestation, adjusted for the presence of dental health aide therapist/dentist in the community, had offspring with higher dmft score/days of life at 12–35 months of age than did those with a sufficient 25(OH)D concentrations (RR 1.3, *p* < 0.0001). Access to dental care, through living in a community with a dental health aide therapist/dentist, was associated with a lower dmft score (RR 0.8, *p* = 0.00012). After removing 99 visits with 0 dmft, children of women with deficient 25(OH)D concentrations had 40% higher dmft scores at 12–35 months than children of women with non-deficient 25(OH)D (RR 1.4, *p* = 0.03). 

### 3.4. Rickets

Three cases of nutritional rickets in YK Delta children <10 years were confirmed by the ANMC pediatric endocrinologist during the period 2001–2017 (in 2017, a YK Delta infant whose mother’s prenatal 25(OH)D was 6 ng/mL was diagnosed with congenital rickets). During the period 2018–2021, there were no further confirmed cases of childhood nutritional rickets in YK Delta children. In addition, the overall incidence of nutritional childhood rickets in Alaska Native children aged <10 years confirmed by the ANMC pediatric endocrinologist decreased by 48% from 4.9 cases per 100,000 children (n = 20) during the period 2001–2016 to 2.5 cases per 100,000 (n = 3) children during the period 2017–2021.

## 4. Discussion

The initiation of routine daily prenatal supplementation with 1000 IU of vitamin D_3_ in a population with high rates of prenatal vitamin D deficiency and childhood rickets resulted in a 66% decrease (from 15% to 5%) in prenatal women with late pregnancy vitamin D deficiency. There may be a decrease in the cases of rickets, but given the relatively few cases, more follow-up years of data are needed. Consistent with other YK Delta studies, 25(OHD) concentrations were higher in summer months, with an offset of 6 weeks [5], and were higher in river communities, which have an abundance of vitamin D-rich salmon and lush fish (burbot). Adherence to vitamin D supplementation as measured by pharmacy refills was associated with higher late-pregnancy 25(OH)D concentrations. Although the etiology of early childhood caries is multi-factorial, vitamin D sufficiency appears to have a role in prevention of ECC.

Prenatal vitamin D supplementation studies show increased maternal vitamin D concentrations at term and in offspring of mothers supplemented with vitamin D compared to placebo or control groups [27,28,29]. Multiple studies indicate that current U.S. prenatal vitamin D supplementation guidelines may not ensure vitamin D levels that are sufficient to prevent congenital or early rickets [30]. Fetal-infant bone mineral accrual is rapid in the third trimester, and routine vitamin D_3_ supplementation of prenatal women at high risk of vitamin D deficiency is one of the main interventions that can reduce the risk of infant rickets by ensuring that infants are born with adequate vitamin D stores. In a study of prenatal women from Toronto, Canada, supplementation with 400 IU/day only increased vitamin D levels in women by 4.2 ng/mL (10.5 nmol/L) between baseline and the third trimester, which is inadequate to bring women out of severe vitamin D deficiency [30]. In contrast, routine supplementation with 1000 IU/day in addition to the 400 IU of vitamin D in prenatal vitamins significantly decreased the proportion of late gestation prenatal women with vitamin D deficiency in the YK Delta. 

Nutritional rickets continues to be diagnosed in Canada, with northern Indigenous infants disproportionately impacted [31]. Canada has also addressed vitamin D supplementation in a northern latitude population. Some variation exists regarding vitamin D intake recommendations for pregnant women in the U.S. vs. Canada. In the U.S., the American College of Obstetricians and Gynecologists recommends prenatal vitamins contain 400 IU of vitamin D but considers 1000–2000 IU/day safe if vitamin D deficiency is identified [32]. In Canada, the Society of Obstetricians and Gynecologists recommends that all women should take a vitamin D supplement if they consume insufficient dietary vitamin D, have darker skin pigmentation, or cover their skin [33], while the Canadian Pediatric Society recommends that up to 2000 IU/day during the winter should be considered for pregnant women. In addition, the Society recommends that Indigenous pregnant women receive routine vitamin D supplementation of 400 IU/day, with an additional 1000 IU/day recommended for those with risk factors for vitamin D deficiency [31]. 

In our study, prenatal women with insufficient plasma 25(OH)D concentrations in late pregnancy had offspring with higher dmft score/days of life at 12–35 months of age than those with sufficient 25(OH)D concentrations. This adds to the evidence for a relationship between low vitamin D levels and ECC likely mediated by impact on enamel formation [19,22,34,35]. Prenatal calcium levels have been shown in a Canadian birth cohort to be inversely associated with enamel hypoplasia in infants [36]. Increased levels of vitamin D may also provide caries-protective benefits. A pilot supplementation study of 100,000 IU vitamin D2 during pregnancy in Winnipeg did not demonstrate an improvement in cord 25(OH)D or dental outcomes for infants [37], but did identify significant inverse association between 25(OH)D level and the number of decayed primary teeth in infants. 

This study has the inherent limitations of an observational study comparing a treatment period to historical controls. These include selection bias, since we only included women who had 25(OH)D measurements, an ecological bias for rickets association, and the limitation of using pharmacy refill data to estimate adherence. The historical 2015–2016 data includes fewer 25(OH)D measurements because they were obtained at providers’ discretion and also routinely at delivery during summer 2016. In contrast, 25(OH)D was routinely obtained at first prenatal and third trimester during the post-supplementation period. However, we found no difference in early pregnancy mean 25(OH)D concentrations between women in the post-supplementation and pre-supplementation groups. We also found that mean 25(OH)D concentrations did not change between early and late pregnancy among the pre-supplementation group, whereas mean 25(OH)D increased significantly between early and late pregnancy for women in the post-supplementation group. The strength of this study was the access to the health records for all prenatal women and the dental records for their infants and the ability to evaluate trends in 25(OH)D in the entire population of prenatal women from this relatively isolated region with a single health care system.

## 5. Conclusions

Our evaluation indicates prenatal vitamin D supplementation with 1000 IU vitamin D_3_ increased prenatal 25(OH)D concentrations substantially and might have contributed to lower rates of ECC in their children. These findings support the Alaska Vitamin D Workgroup’s recommendations and current YKHC prenatal vitamin D supplementation guidelines. These data may inform prenatal vitamin D supplementation policies in other high-risk populations with high rates of rickets.

## Figures and Tables

**Figure 1 nutrients-14-03935-f001:**
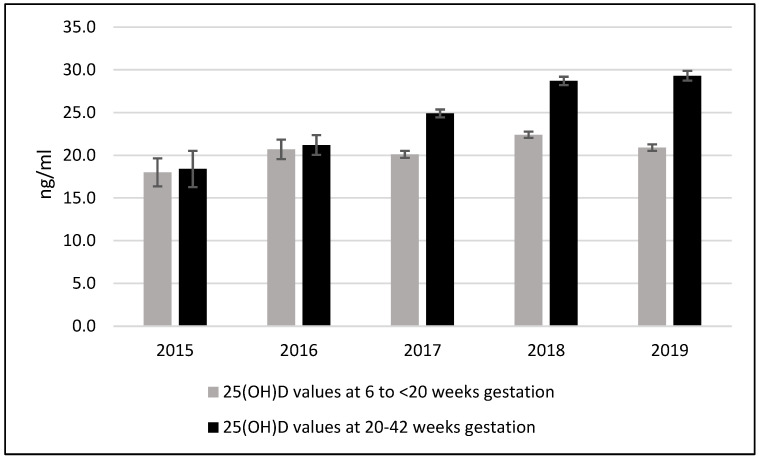
25(OH) vitamin D status pre-supplementation (2015–2016) and post-supplementation (2017–2019) among prenatal women 20+ weeks gestation, Yukon Kuskokwim Delta region. (Deficient <12 ng/mL; Insufficient ≥12 ng/mL and <20 ng/mL; Sufficient ≥20 ng/mL).

**Table 1 nutrients-14-03935-t001:** Characteristics of prenatal women 20+ weeks gestation with vitamin D testing, Yukon Kuskokwim Delta, 2015–2019 (N = 1522).

**Maternal Characteristics, Mean (SD)**	
Maternal age, years	25.9 (5.6)
Gestational age at 20+ weeks, weeks	33.1 (5.9)
Prenatal plasma 25(OH) vitamin D, ng/mL	26.5(5.6)
**Season at prenatal visit, n (%)**	
Nov-Apr	771 (50.7)
May-Oct	751 (49.3)
**Sub-region, n (%)**	
Coastal	682 (44.8)
River	840 (55.2)
**Dental Health Aide Therapist/Dentist in Community, n (%)**	
Yes	627 (41.2)
No	895 (58.8)
**Serum 25-hydroxy vitamin D concentration, n (%)**	
Deficient	94 (6.2)
Insufficient	344 (22.6)
Sufficient	1084 (71.2)

**Table 2 nutrients-14-03935-t002:** Bivariate analysis of late pregnancy (20+ weeks gestation) plasma 25 hydroxyvitamin D concentration in Yukon Kuskokwim Delta prenatal women by maternal characteristics, 2015–2019.

	Mean (SD) 25(OH)D (ng/mL) N = 1522	*p*-Value ^1^
**Maternal Age, years**		
<33	26.3 (10.6)	0.06
≥33	27.8 (10.6)	
**Gestational Age, weeks**		
<36	26.9 (10.4)	0.14
≥36	26.1 (10.8)	
**Season at prenatal visit**		
Nov-Apr	25.9 (11.4)	0.03
May-Oct	27.1 (9.7)	
**Sub-region**		
Coastal	25.5 (11.2)	0.0008
River	27.3 (10.0)	

^1^ Independent *t*-test.

**Table 3 nutrients-14-03935-t003:** Bivariate analysis of prenatal characteristics and 25-hydroxyvitamin D concentrations in late prenatal (20+ weeks gestation) Yukon Kuskokwim Delta women pre-supplementation (2015–2016) versus post-supplementation (2017–2019).

25(OH) Vitamin D Concentration	2015–2016	2017–2019	*p*-Value
	**n = 175**	**n = 1347**	
Mean	20	27.3	<0.0001 ^1^
SD	7.9	10.6	
Min	5.8	5	
Max	41.7	100	
**25 (OH) Vitamin D concentration**	**n (%)**	**n (%)**	** *p* ** **-value**
Deficient <12 ng/mL	26 (14.9)	68 (5.0)	<0.0001 ^2^
Insufficient ≥12 ng/mL and <20 ng/mL	76 (43.4)	268 (19.9)	
Sufficient ≥20 ng/mL	73 (41.7)	1011 (75.1)	
	**Mean (SD)**	**Mean (SD)**	** *p* ** **-value**
**Maternal Age, years (n = 1522)**	**n = 175**	**n = 1347**	
<33 (n = 1315)	19.8 (7.7)	27.2 (10.6)	<0.0001 ^1^
≥33 (n = 207)	21.4 (9.2)	28.6 (10.6)	0.003
**Season at prenatal visit** (n = 1522)			
Nov-Apr (n = 771)	19.8 (8.0)	26.9 (11.6)	<0.0001 ^1^
May-Oct (n = 840	20.4 (7.8)	27.7 (9.6)	<0.0001 ^1^
**Sub-region** (n = 1522)			
Coastal	18.0 (7.1)	26.5 (11.3)	<0.0001 ^1^
River	21.8 (8.2)	28.0 (9.9)	<0.0001 ^1^

^1^ Independent *t*-test. ^2^ Chi square *p*-value.

## Data Availability

Data described in the manuscript, code book, and analytic code will be made available upon request pending approval by the tribal health organizations.

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
