# Peer review of "Impact of a Prenatal Vitamin D Supplementation Program on Vitamin D Deficiency, Rickets and Early Childhood Caries in an Alaska Native Population"

_nutrients, 2022, doi:10.3390/nu14193935_

Round 1
Reviewer 1 Report
The manuscript of the present study presented was aimed to analyze the impact of prenatal Vit-D supplementation program in the neonate/child-bearing native females (in different risk rankings groups based on the {25(OH)-Vit-D} measured) was effective for the outcomes on rickets and ECC in the Alaska YKD region through health e-records and surveys/of-(K-A-B) from YKHC & NMC between 2015-2019 for serum/cord-blood [25(OH)-Vit-D] concentrations, supplement-refills regimens and ECC (1-3-yr-old)/rickets incidences collected in the region-specific study cohorts (of prenatal females of late-pregnancy N-1522: collected from 175 pre-supplementation in YR-2015-2016 vs. 1347 post-supplementation in YR-2017-2019).
The results presented are interesting by design and nature of interpretations; yet, there are some caveats that will require further explanations before it may be considered for publication by the prestigious Nutrients J, as follows:
1) There appeared to have some characteristics issues presented either with bias or somehow discrepancies between Table-1 & Table-3, regarding the overall, averaged [25(OH)-vit-D] concentrations (Table-1: 94, 344, 1084) & the mean values vs. the distributed mean values among all three categories in Bivariate analysis for the pre vs. post-supplementation groups (Table-3: below line 215), requiring some explanations for their add-ups. As this information is needed to address the major changes detected based on the re-distributed [25(OH)-vit-D] concentrations measured among all three sub-groups from pre- to post-supplementations in the study,
2) What was the high-risk population in the present study? The deficient or insufficient group? Or, are both the deficient & insufficient groups together? The authors appeared to include both groups’ results together for the results and interpretations as a whole. Yet, if the latter one, insufficient group, is included herein, further explanation is needed to justify the concern of its borderline issues as to why and how it (the Vit-D supplementation program) was included for the resultant works.
Especially, further to the above point, and in the Conclusion statement in the abstract & summary: Prenatal Vit-D supplementation can improve the childhood outcome in the high-risk group. The authors meant the deficient or insufficient group, which one or both ones, together??
3) What is (or are) the correlation(s) for the outcome measures on ECC & rickets in the present cohort study?? The results presented in the manuscript, though seemingly in and towards the same reduction-trend, cannot be rectified unless more details analyses on both outcome measures are presented and assessed; for instance, multivariate regression analyses, subgrouping analyses … etc., as to the resultant reductions measured in all three groups (i.e., Table-3 & Fig.-25) came from the key-variance that statistically or practically derive the differences detected.
4) In the cohorts’ population characteristics described, what were “the other” environmental factors in the Alaska wild that may be involved and likely be measured in the present study: such as the source of non-piped water, especially other metals exposures that may likely affect the marine/non-marine source on serum nitrogen isotope ratio values?? In addition, the piped/non-piped water “fluoride levels” having been used and the source of these fluorides applied by among the residents in the study regions??
On the same token, the environmental exposure(s) that can significantly affect the ECC outcome on dmft measures; for example, the serum “lead” exposures from the environment or households, which has been shown scientifically in both human and animal studies to affect the ECC incidences and prevalence.
5) The manuscript will need to address, at least describe some concerns about, other critical nutritional aspects and behavioral issues as to how these factors may be involved or/and be disregarded on the ECC & ricket assessments for the outcome measures.
6) Further to the above points in 4) & 5), other “maternal vs. neonate issues & characteristics” need to be mentioned or addressed further, at least; as the low or lower serum [25(OH)-vit-D] concentrations are not the single or the most determinant for ricket vs. ECC per se, and if there are other maternal aspects of the physical or physiological measures, such as maternal BMI index & medications, the neonate’s term-weights or pre-term weights measured, …etc., For the designs of the modern cohort study analyses, these factors require some explorations and further comments in more descriptive details.
7) It is known that the serum levels of serum [25(OH)-vit-D3] concentrations fluctuate on regular basis [whose half-life is about 2-3 weeks in the susceptible hosts, & its dosing requires some 5-6 months to reach its steady state]. Were there any measures/protocols to ensure or rectify the results of the concentrations measured in the study that were close to or the same as real values in the hosts?? Are any parallel measures from the host serum taken for comparisons, rather than a single outcome assessment?? Are any other health records on serum biomarkers available?? Please comment on these.
8) This reviewer is not sure whether the 3 cases of nutritional rickets described in the manuscript is directly related to the present cohort study for its summary or conclusion or NOT, since they were from the 2001-2021 period, which may not be considered useful in the present manuscript. The commentary from the editorial board/editors may be discretive.
Specific comment: The above manuscript present is practically useful on its own for health guidelines and promotions. Yet, some concerns described above may be applied in order to justify the summary and conclusion derived from this interesting cohort for the study of the Alaskan native population.
Author Response
The manuscript of the present study presented was aimed to analyze the impact of prenatal Vit-D supplementation program in the neonate/child-bearing native females (in different risk rankings groups based on the {25(OH)-Vit-D} measured) was effective for the outcomes on rickets and ECC in the Alaska YKD region through health e-records and surveys/of-(K-A-B) from YKHC & NMC between 2015-2019 for serum/cord-blood [25(OH)-Vit-D] concentrations, supplement-refills regimens and ECC (1-3-yr-old)/rickets incidences collected in the region-specific study cohorts (of prenatal females of late-pregnancy N-1522: collected from 175 pre-supplementation in YR-2015-2016 vs. 1347 post-supplementation in YR-2017-2019).
The results presented are interesting by design and nature of interpretations; yet, there are some caveats that will require further explanations before it may be considered for publication by the prestigious Nutrients J, as follows:
1) There appeared to have some characteristics issues presented either with bias or somehow discrepancies between Table-1 & Table-3, regarding the overall, averaged [25(OH)-vit-D] concentrations (Table-1: 94, 344, 1084) & the mean values vs. the distributed mean values among all three categories in Bivariate analysis for the pre vs. post-supplementation groups (Table-3: below line 215), requiring some explanations for their add-ups. As this information is needed to address the major changes detected based on the re-distributed [25(OH)-vit-D] concentrations measured among all three sub-groups from pre- to post-supplementations in the study. Thank you. Table 1 included characteristics of all prenatal women with 25(OH)D testing at 20+ weeks gestation (n=1522). Table 3 includes the same women, divided between pre-supplementation (2015-2016) and post-supplementation (2017-2019). We have added headers above the last 3 categories to indicate n=175 for pre-supplementation and n=1347 during post-supplementation periods.
2) What was the high-risk population in the present study? The deficient or insufficient group? Or, are both the deficient & insufficient groups together? The authors appeared to include both groups’ results together for the results and interpretations as a whole. Yet, if the latter one, insufficient group, is included herein, further explanation is needed to justify the concern of its borderline issues as to why and how it (the Vit-D supplementation program) was included for the resultant works. The “high-risk population” refers to this northern latitude Alaska Native population which experiences a high rate of rickets in children and vitamin D deficiency in prenatal women.
Especially, further to the above point, and in the Conclusion statement in the abstract & summary: Prenatal Vit-D supplementation can improve the childhood outcome in the high-risk group. The authors meant the deficient or insufficient group, which one or both ones, together?? The designation “high risk populations” in the Conclusion refers to populations with high rates of rickets. We added “with high rates of rickets” to the Conclusion of the abstract and manuscript.
3) What is (or are) the correlation(s) for the outcome measures on ECC & rickets in the present cohort study?? The results presented in the manuscript, though seemingly in and towards the same reduction-trend, cannot be rectified unless more details analyses on both outcome measures are presented and assessed; for instance, multivariate regression analyses, subgrouping analyses … etc., as to the resultant reductions measured in all three groups (i.e., Table-3 & Fig.-25) came from the key-variance that statistically or practically derive the differences detected. Our study only examined the relationship between ECC and prenatal Vitamin D values. Information about rickets incidence came from separate statewide electronic medical record queries following methods previously published in reference 3.
4) In the cohorts’ population characteristics described, what were “the other” environmental factors in the Alaska wild that may be involved and likely be measured in the present study: such as the source of non-piped water, especially other metals exposures that may likely affect the marine/non-marine source on serum nitrogen isotope ratio values?? In addition, the piped/non-piped water “fluoride levels” having been used and the source of these fluorides applied by among the residents in the study regions?? The natural levels of fluoride in the non-piped water in YK Delta is low and only the town of Bethel in the YK Delta had fluoridated water.
The use of serum and RBC nitrogen isotope ratios has been validated in this population and other northern latitude populations and has a high correlation with 25(OH)D. Metals do not affect the nitrogen isotope ratio. We did not use that analysis in the current study and thus a discussion of the method is outside the scope of the paper.
On the same token, the environmental exposure(s) that can significantly affect the ECC outcome on dmft measures; for example, the serum “lead” exposures from the environment or households, which has been shown scientifically in both human and animal studies to affect the ECC incidences and prevalence. There are generally low levels of lead in the environment and households. Lead exposure in the environment and homes was not a variable included in this study as that was not an objective of this study.
5) The manuscript will need to address, at least describe some concerns about, other critical nutritional aspects and behavioral issues as to how these factors may be involved or/and be disregarded on the ECC & ricket assessments for the outcome measures. We agree that infant nutrition and feeding practices are significant contributors to ECC and rickets. In the Introduction we added a sentence line 65 “Exposure to sugars, improper infant feeding, inadequate dental hygiene and lack of access to care are known risk factors for early dental decay.” We added information from a previous publication(reference 3) in line 48 “Alaska Native children with rickets/vitamin D deficiency had similar breast-feeding prevalence but were less likely to have received vitamin D supplementation”.
6) Further to the above points in 4) & 5), other “maternal vs. neonate issues & characteristics” need to be mentioned or addressed further, at least; as the low or lower serum [25(OH)-vit-D] concentrations are not the single or the most determinant for ricket vs. ECC per se, and if there are other maternal aspects of the physical or physiological measures, such as maternal BMI index & medications, the neonate’s term-weights or pre-term weights measured, …etc., For the designs of the modern cohort study analyses, these factors require some explorations and further comments in more descriptive details. Thank you. We evaluated maternal parity, age, subregion, tobacco use, birth weight and season. Birthweight was not significantly associated with maternal 25(OH)vit D concentration. We added the analysis of birthweight and maternal 25(OH) D as section 3.2 of the results.
7) It is known that the serum levels of serum [25(OH)-vit-D3] concentrations fluctuate on regular basis [whose half-life is about 2-3 weeks in the susceptible hosts, & its dosing requires some 5-6 months to reach its steady state]. Were there any measures/protocols to ensure or rectify the results of the concentrations measured in the study that were close to or the same as real values in the hosts?? Are any parallel measures from the host serum taken for comparisons, rather than a single outcome assessment?? Are any other health records on serum biomarkers available?? Please comment on these. For women in both the pre-supplementation (2015-2016) and post-supplementation (2017-2019) study groups, we queried the electronic medical record for serum 25(OH)-vit-D3 levels measured during early gestation and late gestation (see Figure). No other measurements were obtained, nor were measurements of biomarkers of vitamin D status. While there could be fluctuation in 25(OH)-vit-D3 levels during pregnancy due to unknown factors, the difference between early and late gestation measurements did not change, on average, in the pre-supplementation period (2015-2016), but increased, on average, in the post-supplementation period (2017-2019). We attribute this non-random change to vitamin D supplementation during pregnancy during the observation period.
8) This reviewer is not sure whether the 3 cases of nutritional rickets described in the manuscript is directly related to the present cohort study for its summary or conclusion or NOT, since they were from the 2001-2021 period, which may not be considered useful in the present manuscript. The commentary from the editorial board/editors may be discretive. Since rickets is an uncommon condition, we included the period of 2001-2021 to reflect the pre- post supplementation period for describing rickets trends in the population. The rickets data is from statewide queries and not part of data collected to examine the relationship between maternal Vitamin D values and ECC.
Specific comment: The above manuscript present is practically useful on its own for health guidelines and promotions. Yet, some concerns described above may be applied in order to justify the summary and conclusion derived from this interesting cohort for the study of the Alaskan native population.

Reviewer 2 Report
This is an observational cohort study that looks at the impact of a prenatal vitamin D supplementation program on the prevalence of maternal and infant vitamin D deficiency determined by 2OH-vitamin D levels, and on the diagnosis of early childhood rickets and childhood caries. The impact on 25OH-vitamin D serum levels is convincing but very unconvincing for rickets. The title of this paper should really be “…..The Impact of a Prenatal Vitamin D Supplementation on Vitamin D Deficiency and Early Childhood Nutritional Rickets and Dental Caries in an Alaskan Native Population”. Much of the US and Canada is at a northern latitude where true vitamin D deficiency and childhood nutritional rickets is less of a concern. Please indicate whether or not informed consent was considered for this study and whether or not an ethics board decided informed consent was not necessary. This is a bit more than a quality improvement study and targets a specific population.
Other comments:
1. Please include a list of abbreviations used in the manuscript.
2. Lines 134-135. Please be more specific about the criteria used to diagnose clinical crickets
3. Line 141: was the vitamin D questionnaire sent to ALL YKHC employees, not just health care providers? Were these individuals also asked to complete the questionnaire themselves, not just provide the questionnaire to mothers?
4. Line 167: replace “outcome” with “dependent variable”.
5. Table 2. The mean 25OHD value (27.3) is left out for “River”
in Table 3.
6. The Figure is incorrectly labeled as Figure #25. Is this figure #1? This figure is not directly referred to in the results section and just duplicates what is in the tables. Is it necessary?
7. Lines 222-228. What was the actual compliance for the number of refills? The number of mothers who did not comply is not indicated here.
8. Lines 230-238 The survey data is probably not meaningful and highly biased based on about a 2% response rate. This reviewer would advise leaving this out of the study. We do not know how many surveys were sent out to health care providers of YKHC employees. At most, this could be supplementary information.
9. Lines 240-247. What does “0” dfmt score mean? Does this mean the data was not collected, or that the actual score was zero? Does it mean there were no dental visits in the medical record for a given child? It would be informative if data from which the RR scores for dfmt are tabulated, was presented in a table format. The authors are encouraged to do this. How many of eligible infants had dental visits recorded and quantifiable dfmt scores? Presumably, infants who did not have access to dental visits were not included? How many infants had no access? How many children had access to dental care but did not take advantage of it?
1O. Lines 249-256. The number of cases of rickets overall is very small and no clinical difference can really be determined between the pre and post supplementation periods. Whether supplements had any impact on the prevalence or incidence of rickets is questionable. In Line 253 please indicate that this is “nutritional” rickets.
10. Line 261. Leave out “apparent risk of rickets.” The observations should be more nuanced, something like : There may be a decrease in the cases of rickets, but given the relatively few cases, more follow-up years of data are needed.
11. Lines 266-267. Leave out reference to the survey data.
12. Lines 304-308. Was the vitamin D2 given intramuscularly in a single dose? Would the use of vitamin D3have made any difference?
13. Lines 312-316. These lines are confusing, and particularly the phrase “and routinely in the summer of 2016”. Why were they routineca in the summer of 2016 but not summer of 2015?
14. Lined 327-228. No rate of rickets was calculated. At most “may possibly have decreased the number of cases of rickets but longer followup is necessary.”
Author Response
This is an observational cohort study that looks at the impact of a prenatal vitamin D supplementation program on the prevalence of maternal and infant vitamin D deficiency determined by 2OH-vitamin D levels, and on the diagnosis of early childhood rickets and childhood caries. The impact on 25OH-vitamin D serum levels is convincing but very unconvincing for rickets. The title of this paper should really be “…..The Impact of a Prenatal Vitamin D Supplementation on Vitamin D Deficiency and Early Childhood Nutritional Rickets and Dental Caries in an Alaskan Native Population”. Thank you. We have changed the title to “Impact of a Prenatal Vitamin D Supplementation Program on Vitamin D Deficiency, Rickets and Early Childhood Caries in an Alaska Native Population”
Much of the US and Canada is at a northern latitude where true vitamin D deficiency and childhood nutritional rickets is less of a concern. Vitamin D deficiency rickets and other manifestations of severe vitamin D deficiency continue to be seen in Canada, particularly in northern indigenous children, prompting a recent update in recommendations by the Canadian Pediatric Society. (Irvine JW, Ward LM Canadian Paediatric Society; First Nations, Inuit, and Metis Health Committee. Preventing symptomatic vitamin D deficiency and rickets among Indigenous infants and children in Canada. Paediatrics & Child Health 2022, In press” We added a sentence in the Discussion, line 297, “Nutritional rickets continues to be diagnosed in Canada, with northern Indigenous infants disproportionately impacted.
Please indicate whether or not informed consent was considered for this study and whether or not an ethics board decided informed consent was not necessary. As described in line 114-116 we received Alaska Area IRB approval and ethics approval from tribal health organization with a waiver of informed consent.
This is a bit more than a quality improvement study and targets a specific population.
Other comments:
- Please include a list of abbreviations used in the manuscript.
United States(U.S.)
25-hydroxyvitamin D 25 (OH)D
Early childhood caries (ECC)
Yukon Kuskokwim Health Corporation (YKHC)
Alaska Native Medical Center (ANMC)
International classification of diseases (ICD)
- Lines 134-135. Please be more specific about the criteria used to diagnose clinical rickets
We added the following” “the pediatric endocrinologist evaluated the clinical and laboratory findings for potential cases of rickets (ICD9 268.0, 268.1 or ICD10 E55.0) identified in the electronic medical record as previously published. Children were included as having nutritional rickets if they had clinical/radiologic evidence of rickets, not associated with underlying malabsorption or hepatic disease.
- Line 141: was the vitamin D questionnaire sent to ALL YKHC employees, not just health care providers? Were these individuals also asked to complete the questionnaire themselves, not just provide the questionnaire to mothers?
Yes, the survey was sent to all YKHC employees to complete themselves. The survey was also provided to pregnant women in prenatal clinic.
- Line 167: replace “outcome” with “dependent variable”. Done.
- Table 2. The mean 25OHD value (27.3) is left out for “River” in Table 3.
Thank you. We added that value which was inadvertently deleted back in Table 2.
- The Figure is incorrectly labeled as Figure #25. Is this figure #1? This figure is not directly referred to in the results section and just duplicates what is in the tables. Is it necessary?
We have removed the incorrect label of “25” to the Figure. This figure identified the trend in increasing late pregnancy 25(OH)D concentrations post-supplementation without a change in early pregnancy 25(OH)D. The figure is referenced on line 200.
- Lines 222-228. What was the actual compliance for the number of refills? The number of mothers who did not comply is not indicated here. Refill data was reviewed for 351 women in 2018, among whom 106 had 0 refills, 53 had 1 refill (30 days), 39 had 2 refills (60 days), 99 had 3 refills (90 days), and 54 had 4 or more refills (>120 days). We added n=106 for women with 0 refills and n=54 for women with 4 or more refills.
- Lines 230-238 The survey data is probably not meaningful and highly biased based on about a 2% response rate. This reviewer would advise leaving this out of the study. We do not know how many surveys were sent out to health care providers of YKHC employees. At most, this could be supplementary information.
We agree that the survey has a small response rate; however, it provides some confirmation of the acceptability of prenatal supplementation among community members. We will remove this from the manuscript.
- Lines 240-247. What does “0” dmft score mean? Does this mean the data was not collected, or that the actual score was zero?Does it mean there were no dental visits in the medical record for a given child? No, we only included dmft scores for children who had received a comprehensive dental examination. A dmft score of “0” means that there were no decayed missing or filled teeth and the actual score was zero. We excluded those with dmft scores of 0 in order to assess whether maternal vitamin D levels were associated with numbers of decayed teeth.
It would be informative if data from which the RR scores for dmft are tabulated, was presented in a table format. The authors are encouraged to do this. How many of eligible infants had dental visits recorded and quantifiable dmft scores? Presumably, infants who did not have access to dental visits were not included? How many infants had no access? How many children had access to dental care but did not take advantage of it? All infants had access to dental care, however, many parents only sought dental care when a child had pain from decay.
1O. Lines 249-256. The number of cases of rickets overall is very small and no clinical difference can really be determined between the pre and post supplementation periods. Whether supplements had any impact on the prevalence or incidence of rickets is questionable. In Line 253 please indicate that this is “nutritional” rickets. We agree that the number of cases of rickets is small; however, it represents a rate of rickets that is higher than in the general U.S. population. Because of the small number of cases, it will take several years to demonstrate a significant change, but we believe that the information on trends in nutritional rickets are informative.
- Line 261. Leave out “apparent risk of rickets.” The observations should be more nuanced, something like : There may be a decrease in the cases of rickets, but given the relatively few cases, more follow-up years of data are needed. We have replaced “apparent decrease in rickets” with this more nuanced language.
- Lines 266-267. Leave out reference to the survey data. We removed reference to the survey data.
- Lines 304-308. Was the vitamin D2given intramuscularly in a single dose? Would the use of vitamin D3 have made any difference? This Canadian study involved oral supplementation with doses of 50,000 IU vitamin D2. That study did discuss in the limitations section that supplementation with 100,000 IU vitamin D3 might have resulted in improved 25(OH)D concentrations. It might also have resulted in lower caries scores in the supplemented group.
- Lines 312-316. These lines are confusing, and particularly the phrase “and routinely in the summer of 2016”. Why were they routinely in the summer of 2016 but not summer of 2015? Prior to the supplementation guidelines, 25(OH)D testing was performed at the providers’ discretion; however, YKHC evaluated 25(OH)D routinely in women at delivery during summer 2016 (see lines 83-84 of Background). We revised the wording to “ The historical 2015-2016 data includes fewer 25(OH)D measurements because they were obtained at providers’ discretion and also routinely at delivery during summer 2016.”
- Lined 327-228. No rate of rickets was calculated. At most “may possibly have decreased the number of cases of rickets but longer follow-up is necessary.” We removed “nutritional rickets” from line 329.
Round 2
Reviewer 1 Report
In the amended manuscript provided, the authors have intelligently revised and updated most of the reviewer's comments with a much more focused description and in its contents for appropriately justified interpretations.